# Properties and Applications of Graphene and Its Derivatives in Biosensors for Cancer Detection: A Comprehensive Review

**DOI:** 10.3390/bios12050269

**Published:** 2022-04-24

**Authors:** Mehrab Pourmadadi, Homayoon Soleimani Dinani, Fatemeh Saeidi Tabar, Kajal Khassi, Sajjad Janfaza, Nishat Tasnim, Mina Hoorfar

**Affiliations:** 1School of Chemical Engineering, College of Engineering, University of Tehran, Tehran 1417935840, Iran; mehrabpourmadadi@gmail.com (M.P.); fateme.saedi@ut.ac.ir (F.S.T.); 2Department of Electrical and Computer Engineering, Missouri University of Science and Technology, Rolla, MO 65409, USA; hshnm@umsystem.edu; 3Department of Textile Engineering, Isfahan University of Technology, Isfahan 8415683111, Iran; k.khassi@alumni.iut.ac.ir; 4School of Engineering, University of British Columbia, Kelowna, BC V1V 1V7, Canada; janfaza@ualberta.ca (S.J.); nishat.tasnim@ubc.ca (N.T.); 5School of Engineering and Computer Science, University of Victoria, Victoria, BC V8W 2Y2, Canada

**Keywords:** graphene, cancer biomarkers, nanobiosensor, biosensing method, diagnostic applications

## Abstract

Cancer is one of the deadliest diseases worldwide, and there is a critical need for diagnostic platforms for applications in early cancer detection. The diagnosis of cancer can be made by identifying abnormal cell characteristics such as functional changes, a number of vital proteins in the body, abnormal genetic mutations and structural changes, and so on. Identifying biomarker candidates such as DNA, RNA, mRNA, aptamers, metabolomic biomolecules, enzymes, and proteins is one of the most important challenges. In order to eliminate such challenges, emerging biomarkers can be identified by designing a suitable biosensor. One of the most powerful technologies in development is biosensor technology based on nanostructures. Recently, graphene and its derivatives have been used for diverse diagnostic and therapeutic approaches. Graphene-based biosensors have exhibited significant performance with excellent sensitivity, selectivity, stability, and a wide detection range. In this review, the principle of technology, advances, and challenges in graphene-based biosensors such as field-effect transistors (FET), fluorescence sensors, SPR biosensors, and electrochemical biosensors to detect different cancer cells is systematically discussed. Additionally, we provide an outlook on the properties, applications, and challenges of graphene and its derivatives, such as Graphene Oxide (GO), Reduced Graphene Oxide (RGO), and Graphene Quantum Dots (GQDs), in early cancer detection by nanobiosensors.

## 1. Introduction

Cancer is the leading cause of mortality worldwide [1,2]. Cancer occurs when normal cells in a certain part of an organ begin to grow out of control. Cancer cells can invade adjacent tissues and finally spread to other regions of the organ. In general, cancer cells are produced from normal cells due to unrepairable DNA damage [3]. It is estimated that the world’s population will reach 8.3 billion by 2025, of which more than 20 million modern cases of cancer will be reported each year [4]. Diagnosing cancer in its early stages and providing prompt and appropriate treatment are essential elements in cancer control [5]. Early detection of cancer means the detection of tumors in the early stages of its development, and therefore it is expected that with this strategy, the process of recovery will advance [6]. By the time symptoms become visible, cancer may have begun to expand and be harder to treat. Several screening tests have been shown to detect cancer early and reduce mortality. The United States National Cancer Institute (NCI) defines a biomarker as “a biological molecule found in blood, other body fluids, or tissues that is a sign of a normal or abnormal process or of a condition or disease. A biomarker may be used to see how well the body responds to a treatment for a disease or condition”. In the early stages of cancer, there is only a low level of biomarkers. For this reason, ideally, biomarker identification should be conducted with high sensitivity. Genomic and proteomic detection methods such as polymerase chain reaction (PCR), Southern blot, DNA sequencing, liquid chromatography mass spectrometry (LC-MS), and immunoassay techniques such as enzyme-linked immunosorbent assay (ELISA) are also used to diagnose cancer. These tests are not sensitive enough to detect low concentrations of biomarkers in the early stages of cancer, so false-negative results can be obtained, and also these methods are very expensive and technically complex for routine clinical diagnoses [7]. Other methods of diagnosing cancer, such as CT scans, imaging, laboratory tests, tumor biopsy, endoscopic examination, and surgery, can be used. For example, CT scans can examine tissues inside the body and their shape and improve diagnostic decisions by extracting 3D imaging features that provide important information for diagnosing possible diseases, but show poor soft-tissue contrast resolution, and their ability to distinguish between benign and malignant lesions is limited [8]. However, these common standard methods for diagnosing and treating cancer are costly, time-consuming, and require advanced facilities [6]. The importance of human health has led to the introduction of many clinical tests to diagnose cancers, and there is a growing need to develop more sensitive, accurate, faster, and cheaper tests. Challenges in detecting cancer biomarkers, such as the diversity of cancers and the limited ability of a biomarker to detect all cancers of specific organs with high specificity, have encouraged researchers to make great efforts to identify biomarkers without biopsy. In this regard, biosensors are a very promising tool for the possibility of sensitive, specific, and non-invasive diagnosis for early detection of cancer. Effective, accurate methods of cancer detection and clinical diagnosis are urgently needed. Biosensors are devices that are designed to detect a specific biological analyte by essentially converting a biological entity (i.e., protein, DNA, RNA) into an electrical signal that can be detected and analyzed [7]. 

Biosensor technology is a growing field to satisfy the need for sensitive and rapid detection problems [9]. In addition to the use of commercial SPR-based biosensors using the Kerchman plasma excitation scheme [10], in recent years, the use of optical, electrochemical, piezoelectric, and other types of biosensors has shown successful results in low concentrations of biomarkers for the detection of cancers including breast cancer, lung cancer, and prostate cancer [11]. In fact, with the proliferation of biosensors capable of detecting cancer early, a major revolution has taken place in the field of cancer diagnosis [12]. Biosensors are a set of components that record a physical, chemical, or biological change and convert it into a measurable marking [13]. The main parts of biosensors include the bioreceptor, transducer, and read-out system [14]. The bioreceptor reacts with the target substance and is usually composed of an antibody, aptamer, cell, or enzyme. Another essential part is the transducer, which must convert the biological signal into a measurable signal. The transducer can be electrochemical, optical, or mechanical [15]. Finally, the read-out system displays the final response and reads the measured signal [14,15]. Electrochemical biosensors provide high sensitivity and selectivity for vital biomarkers that are responsible for vital molecular events in tumor formation and progression [16]. In fact, in addition to differentiating tumor cells from normal cells, these sensors lead to the targeted diagnosis of localized tumor cells and circulating tumor cells. Electrochemical sensors are suitable tools for cell counting, cell classification, and the detection of tumor cells. In the diagnosis of tumor cells, electrochemical biosensors achieve not only high sensitivity (limit of detection of 10 tumor cells in a 250 μL sample) and high specificity but also diagnose duplicate antigens at the tumor cell surface and successfully prevent false-positive results [17].

In electrochemical biosensors, a redox reaction takes place between the bioreceptor and the target molecule [18]. The reaction in the electrochemical transducer requires a reference, counter, and working electrode, the working electrode acts as a transducer [19], and tumor antigens are used as biomarkers for cancer diagnosis in biosensors [20]. The ultimate goal of biosensors is to detect signals specific to each disease and cancer, and recently nanoparticles have been widely used in the design of biosensors. These nanoparticles can react with the analyte in the sample and increase the detection range [21,22]. Nanostructured materials as guiding elements in biosensors that increase the stability of bioreceptor loading at the electrode surface by providing a more electroactive surface using functional groups. There are various purposes for using nanoparticles to make nanobiosensors, including increasing the surface to stabilize biomaterials and thus increasing sensitivity, catalyzing the process, large dynamic range, the possibility of reacting at low potentials, and helping the quick transfer of electrons from the active center of the electrode surface reaction in the electrochemical nanobiosensors [23]. Nanoparticles can be easily fabricated using chemical methods and react with the analyte in the sample, and their inherent properties, such as optical or magnetic properties, can be used [24,25]. Graphene and its derivatives are nanoparticles with unique properties and have many applications in nanobiosensors [26]. Graphene is a two-dimensional (2D) sheet of carbon atoms in a hexagonal (honeycomb) configuration that is an allotrope of carbon [27,28]. Graphene sheets are formed by placing carbon atoms side by side, and in a graphene sheet, each carbon atom is bonded to three other carbon atoms [29]. These three bonds are on the same sheet, and the angles between them are equal to 120°. The length of the C-C bond in graphene is about 0.142 nm [30]. The inherent strength of graphene layers results from these bonds, known as covalent bonds [31]. While the carbon bonds are sp^2^ hybridized, and the σ C-C bond is the strongest bond in materials, the pi (π) bond is responsible for the electron conduction of graphene [32]. With these unique structural properties, graphene has been shown to have special properties and has attracted much interest in scientific research [33]. Graphene transfers heat better than any alternative material and is an excellent electrical conductor with unique optical properties. The derivatives of graphene such as Graphene Oxide (GO), Reduced Graphene Oxide (RGO), and Graphene Quantum Dots (GQDs) have entered the field of graphene research and nanobiosensors because of their unique properties [33]. GO is a derivative of graphene that is obtained by graphite using oxidizing materials [34]. GO is one typical 2D structured and oxygenated planer molecular material. RGO is a reduced form of GO that contains a π-conjugated system. The only major difference between GO and RGO is the number of oxygen atoms present and their conductivity [35]. RGO has low electrical conductivity due to the disruption of the main structure of graphene, but it has a high ability to be functionalized with a variety of chemical and biological molecules. GQDs consist of one or a few layers of graphite and are smaller than 100 nm [36]. GQDs have been studied in recent years because of their unparalleled structure and properties, and their dimensions are about 1 to 10 nanometers [37]. Much of the research mentioned in this review is based on technical and technological developments that cover a wide range of topics and many variables. In order to use this basic information in discovering biomarkers for disease prevention and clinical use, comprehensive studies are needed as the first step in marker and biosensor development research. It will be possible exclusively through specific studies on the technological variety of biosensors to control possible systemic and random errors to perform their evaluation in clinical use. Therefore, the biosensors reported in this study are considered as proposed biosensors that should be targeted in future studies rather than biosensors that have matured for use in clinical practice. 

## 2. Graphene-Based Materials and Their Properties

The physical properties of carbon materials are related to their hybridization stats (sp, sp^2^, sp^3^) [38,39,40,41]. Graphene with sp^2^ hybridization is a fine and zero bandgap semiconductor, but the sp^3^ hybridization diamond is a hard insulator. Graphene is highly thermally conductive, chemically stable, and flexible [42]. One of the superior properties of graphene is that its charge carriers treat as massless particles, and they can move with little scattering in an ambient condition [43]. The charge transport of graphene and its electronic properties are due to its great electronic band structure [44]. Especially among all nanomaterials, graphene has a wide surface area (2630 m^2^g^−1^) [32,33] and is available for direct interaction with many biomolecules [45]. It can be wielded with structural defects using low-cost fabrication methods by chemical modification [46]. The sensitivity of the electrical resistance of graphene to the adsorption makes it highly useful for highly sensitive sensing applications [42].

Other unique properties of graphene, including its optical, magnetic, and high elasticity, make it a suitable monolayer structure for preparing several graphene-based nanocomposites. Graphene has one of the highest tensile strengths of all materials and a high Young’s modulus, which is the relation between stress and strain, that gives graphene its great mechanical properties [47,48]. Using graphene in several sensing applications based on an electrochemical read-out has been offered in various chemical and biological sensors [49]. Graphene derivatives, such as GO and RGO, have been consumed for the fabrication of a diverse range of graphene-based nanocomposites in biosensors by a mixture of metal and biomolecules with enhanced sensitivity [42,50]. Because of the great surface-to-volume ratio and functional chemical groups, GO has broad capacities for the adsorption of biomolecules [51]. GO is composed of graphene layers with active oxygen-containing functional groups on its surface, such as hydroxyl, epoxy, and carboxyl. GO has unique physicochemical properties such as its small size between 20 nm–100 nm, conductivity, and optical and electronic properties [52,53]. Furthermore, GO is hydrophilic and water soluble, whereas graphene is hydrophobic and does not dissolve easily in water [33]. GO has a significantly disrupted sp^2^ carbon network and shows many defects, and functional groups make its insulators [32]. GO is not conductive and has reduced mechanical properties compared to graphene. Accordingly, to improve the conductivity of GO, it is necessary to convert GO into RGO. Graphene nanomaterials can be successfully functionalized through non-covalent or covalent interactions. Typical covalent reactions include oxidation, reduction, radicals, and nucleophilic/electrophilic additives(Figure 1) [54]. RGO is produced by removing the functional groups from GO, which partly restores the mechanical and electrical conductivity properties of the graphene layers, and reduction is a process in which sp^3^ carbons are converted to sp^2^ carbons. The electrical, thermal, and mechanical properties of RGO and graphene are similar [55]. RGO exhibits excellent electrochemical behavior such as lower oxidation potential, making it a promising candidate in the fabrication of biosensors. The cost-effectiveness and the controllability of the O_2_ functional groups make RGO crafty for biological applications [56]. RGO has been explored in the fabrication of electrochemical biosensors because its defects and chemical groups simplify charge transfer [57,58,59]. GQDs are known as a new type of zero-dimensional (0D) fluorescent nanomaterial [60]. GQDs expose terrific optoelectronic properties and excellent biocompatibility [61]. These GQDs are superior in chemical inertness, simplicity of production, and low cytotoxicity, and for this reason are appropriate in biosensors [13]. GQDs have the carboxyl and hydroxyl groups at their plane edge, allowing them to expose high water solubility [62].

## 3. Graphene-Based Materials Synthesis Methods

Usual sensing methods are expensive, require high-accuracy equipment and costly reagents, and the majority of reactions are not quantifiable in real-time. Graphene-based sensors are now being used as another method for the identification of cancer-related biomolecules. Graphene and its derivatives must be produced at a low cost to fabricate biosensors for early cancer detection successfully. The synthesis of graphene can be applied in two main ways: top-down (destruction) and bottom-up (construction) methods [63]. Most of the top-down approaches are very scalable and produce high-quality products. These methods can convert major precursors such as graphite and other carbon-based raw materials to nano-sized graphene, and in this method, graphite is used directly as a raw material for the synthesis of graphene. The bottom-up methods synthesize graphene and its derivatives using other carbon sources instead of graphite. In these methods, carbon compounds in the phase of gas, liquid, or solid are used [64]. The bottom-up methods make graphene-based materials with vast surface area and defect-free sheets [65]. An overview of graphene synthesis methods is shown in the Figure 2.

### 3.1. Top-Down Methods

The top-down approaches such as arc discharge [66], oxidative exfoliation reduction [67], mechanical exfoliation [68], unzipping of carbon nanotubes (CNT) [69], and liquid-phase exfoliation (LPE) [70,71] delaminate the layer of graphite to a single-, bi- and few-layer graphene (An example is given in Figure 3).

As shown in Figure 4, in the arc discharge method the reaction container consists of a carbon precursor and graphite bar that are submerged in liquid media or sometimes in a gas, and the electrical current produces a great temperature plasma up to 3727–5727 °C [65]. Arc discharge could produce graphene at an affordable cost. In this method, a carbon precursor is an anode, and the graphite bar is the cathode. Graphene can be synthesized by discharging through an electric arc between the two electrodes. IN addition, the oxidative exfoliation reduction method is used for the synthesis of GO or RGO. There are four primary paths for GO synthesis: Hofmann, Brodie, Hummer, and Staudenmaier [73,74]. For the synthesis of GO by Hummer’s method [75], first, 1 g of graphite powder was appended to 20 mL of H_2_SO_4_. The solution was put in the ice bath and stirred for a few minutes on the stirrer. Then, 3 g of KMnO_4_ was added to the mentioned solution to turn the color of the solution green. After 5 min, 50 mL of distilled water was added dropwise to the obtained solution, and after 10 min, 100 mL of distilled water was added. Finally, 35 mL of H_2_O_2_ was appended to the solution and stirred for 24 h to synthesize GO well. As always, process safety and environmental cost need to be considered during process scale-up. Recently, the Hummer’s method has been used for the synthesis of GO because it is a fast and safe process [75].

### 3.2. Bottom-Up Methods

The bottom-up approaches include chemical vapor deposition (CVD) [64], Substrate-Free Gas-Phase synthesis (SFGP) [77], template route [78], and total organic synthesis. The chemical vapor deposition (CVD) method is a well-known method for the production of thin films and nanoparticles. CVD decomposes hydrocarbon gases at a high temperature (650–1000 °C). These hydrocarbon gases can be methane (CH_4_), acetylene (C_2_H_2_), ethylene (C_2_H_4_), and hexane (C_6_H_14_), to grow graphene sheets on metallic catalysts such as Cu and Ni [72,79]. The carbon precursor separates free carbon and atoms of hydrogen by contacting the hot surface of the metal catalyst. Then, the carbon atoms spread over the surface [80,81]. Methane gas decomposes its components by passing through a metal scaffold (such as a Ni plate) at high temperatures to gradually form a graphene film on the metal scaffold. Using HCL, a scratch is made on the platform to separate the graphene deposit from the Ni scaffold [82,83]. Alternatively, techniques for the synthesis of graphene by freeze-drying during the chemical vapor deposition method are used. Then, through compensating by passing CO_2_, the deposition of GO is formed on the metal plate and at the end reduced to graphene. Graphene deposition scratches from metal plates are separated by HCL. In another method called epitaxial growth, utilizing carbon solubility in various substrates is a method used to produce graphene layers. The process begins by giving the substrate intense heat, and it allows the carbon to dissolve. This heat depends on the properties of the substrate. The carbon source can be added to the substrate using hydrocarbon molecules. The whole sample is then slowly cooled, reducing the solubility of carbon in growth, resulting in carbon atoms separating from the mass of matter and forming graphene islands on its surface. Different metals and compounds such as Ruthenium (Ru), nickel (Ni), and silicon carbide (SiC) can be used in this process [84,85]. Graphene can be prepared by thermal decomposition at 1200–1600 °C of SiC under vacuum [86]. SiC was used under a high-temperature sublimation and remained on the surface of the particles. By controlling the growth conditions, carbon atoms can be arranged to form a graphene layer. As the growth process progresses, new graphene layers are formed between the current graphene layers and the SiC surface; subsequently, the second layer is formed below the first layer. Therefore, the formation of graphene is highly dependent on the structure of the SiC surface. The most common SiC structures used for this method are 4H-SiC and 6H-SiC [87,88]. In substrate-free gas phase (SFGP), that is a novel way for the synthesis of graphene by gas-phase reaction sans the presence of substrates [77]. In this way, a mixture of liquid ethanol and Argon (Ar) gas is transferred to a microwave-produced plasma, and graphene is organized when the ethanol droplets are evaporated and separated in the plasma region [89]. From 164 mg·min^−1^ of ethanol, 2 mg·min^−1^ was produced [90]. However, this method does not have general parametric studies, and more research is needed. This method provides clean and high-quality graphene [90].

## 4. Modification of Graphene by Different Types of Biorecognition Elements

Biosensors are diagnostic tools defined as having a biorecognition element for analyte specificity and a transducer for a measurable signal. There are various measurement methods for measuring biomarkers, and electrochemical measurement methods can be divided into the following categories:

1. Amperometry, the most widely used technique for detecting cancer cells and cancer biomarkers, continuously measures the current resulting from the oxidation or reduction of an electrical species in a biochemical reaction.

2. Chronoamperometry, in which a square wave potential is applied to the working electrode and a steady-state current is measured as a function of time. Potentiometers determine the difference in electrical potential between two electrodes when the cell current is zero. Potentiometric measurements are made through the Nernst equation, the relationship between concentration and potential. In fact, in this method, the measurement of potential is performed as a function of time in response to a constant current or a square wave.

3. Voltammetry is an electro-analytical method that measures current through analytical potential change information. Due to the different methods of measuring potential change, there are also different voltammetric methods, such as cyclic voltammetry (CV), reciprocating voltammetry, linear sweep voltammetry (LSV), differential pulse voltammetry (DPV), square wave voltammetry (SWV), and adsorptive stripping voltammetry (AdSV). Among the various voltammetric methods, SWV and DPV are often used due to their high sensitivity [91].

Many biorecognition elements have been used, ranging from naturally occurring to synthetic constructs (Figure 5). Naturally occurring biorecognition elements, such as antibodies, nucleic acids, and enzymes, are biologically derived constructs that take the benefit of naturally evolved physiological interactions to attain analyte specificity. Synthetic biorecognition elements are artificially engineered structures extended to imitate physiologically defined interactions. 

The limit of detection (LOD) is commonly used as evidence of the quality of a biosensor. The limit of detection is expressed in units of concentration and, following the IUPAC definition, indicates the smallest solute concentration that a given analytical system is able to distinguish with reasonable reliability of a sample without an analyte. The biorecognition element describes both the selectivity and the sensitivity of diagnostic devices. Between the nanomaterials worked for biosensor fabrication, graphene and its derivatives have been displaying the most promise, since they present an enhanced signal response in a variety of sensing usages [80,81]. In addition, graphene-based nanomaterials have a high surface area and excellent biocompatibility with certain types of biomolecules, such as antibodies, enzymes, DNA, and cells [92]. In the following, eminent biorecognition elements are briefly summarized to serve as a deputy of each category. Nucleic acids and antibodies are more or less used as biorecognition elements to detect cancer or their biomarkers [91]. Cancer biomarkers are biological molecules produced by the body or tumor in a person with cancer. Biomarker testing helps characterize alterations in the tumor. Biomarkers can be DNA, RNA, protein, or metabolomic profiles that are specific to the tumor. Therefore, a biomarker is extremely important for the early detection and treatment of cancer. DNA–graphene hybrids are mainly manufactured by the ultrasonication-driven self-assembly process [93]. P. Abdul Rasheed et al. [94] have manufactured a graphene-based electrochemical DNA biosensor for femtomolar detection of the breast cancer-related BRCA1 gene. Aptamers and various types of receptors, such as enzymes and antibodies, and other biorecognition elements can be immobilized through covalent and noncovalent bonds on the graphene surface. Immobilization is one of the most complex phases in the fabrication of a sensor, and the election of the adequate method for immobilization belongs to the physicochemical conditions of the transducers and receptors [95]. Multiple strategies have been extended to immobilize enzymes onto graphene surfaces to make enzyme-based biosensors. The most common methods are mixing, ultrasound, and cyclic voltammetry. These manners allow the adhesion of the enzymes by adsorption, covalent bonding, or physical entrapment [96], and adsorption belongs to physical immobilization and is often used for sensors that have enzyme receptors. The next method is chemical immobilization, which is generally based on creating a chemical bond amongst the functional groups on the surface of the transducers and the receptors. It usually happens via cross-linking chemical reagents such as hexamethylenediamine, glutaraldehyde, etc. Cross-linking is part of the covalent binding that results in strong, highly stable, and impressive binding. Moreover, graphene can prepare a charged region for the adsorption of any charged molecules or metal ions as an interaction in an empty defect. The functionalized region of graphene is capable of directly binding to nanoparticles, enzymes, antigens, antibodies, DNA, and other particular molecules [97]. In general, the purpose of using different types of biorecognition elements is to detect specific molecules, and the types of these biorecognition elements were mentioned in the previous sections. Using these biorecognition elements, the surface of graphene is modified so that they can detect it by connecting to an analyte or target. Functionalized graphene is also more efficient because by functionalizing graphene, the modification is improved using biorecognition elements and leads to better target detection. The types of biorecognition elements used in graphene-based biosensors are shown in Table 1.

**Figure 5 biosensors-12-00269-f005:**
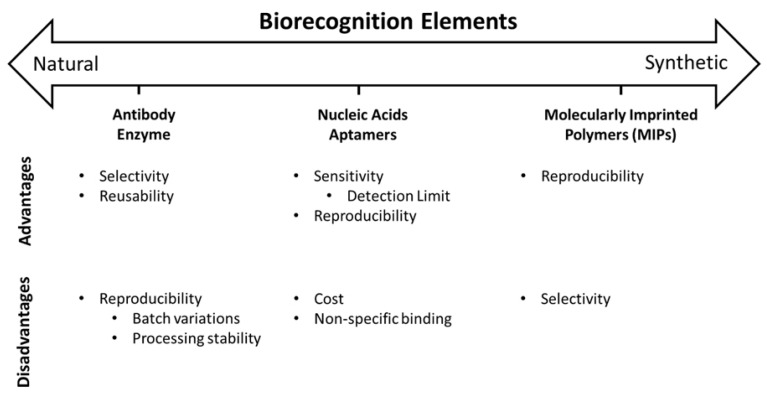
Natural and synthetic biorecognition elements. Reprinted with permission from ref. [97]. Copyright 2018, American Chemical Society.

**Table 1 biosensors-12-00269-t001:** Types of biorecognition elements used in graphene-based biosensors for the detection of various cancer biomarkers.

Biorecognition Element	Sensor Type	Design Method	Biomarker	LOD	Linear Range	Ref.
Aptamer	Electrochemistry	GE/RGONs/Rh-NPs	HER2-ECD	0.667 ng/mL	10.0–500.0 ng/mL	[98]
Antibody	Electrochemistry	COOH-AgPtPd/NH_2_-RGO	PSA	4 fg mL^−1^	4 fg mL^−1^ to 300 ng mL^−1^	[99]
Antibody	Electrochemistry	Graphene-poly (3-aminobenzoic acid)	PSA	0.13 pg	0.01–80 ng/mL	[100]
Antibody	Electrochemistry	RGO	PSA	2 pg/mL 0.06 ng/mL	1–36 ng/mL 0.0018–41.15 ng/mL	[101]
Aptamer	Electrochemistry	GO-carbon nanotubes –hemin	CEA	0.82 fg/mL	1 fg/mL–10 μg/mL	[102]
Antibody	SPR	GO	Cytokeratin 19	1 fg/mL	1 fg/mL^−1^ ng/mL	[103]
Antibody	SERS	GO-AgNPs	PSA	0.23 pg/mL	0.5–500 pg/mL	[104]
Antibody	Electrochemistry	Ag-RGO/CysA-AuNPs	CA15-3	15 U·mL^−1^	15–125 U/mL	[105]
cell	Electrochemistry	Ag-TiO_2_/RGO	CEA	20.5 fg·mL^−1^	-	[106]
Aptamer	Electrochemistry	Amino FG-THI-AuNPs	CEA	2 pg·mL^−1^	0.01–500 ng mL^−1^	[107]
ss DNA	Electrochemistry	NFG/AgNPs/PANI	miRNA-21	0.2 fmol·L^−1^	10 fM–10 µM	[108]
Aptamer	Electrochemistry	RGO/Au TiO_2_/CQDs	PSA	0.007 ng mL^−1^	0.5–7 ng mL^−1^	[109]

## 5. Graphene-Based Biosensors

### 5.1. Graphene-Based FET Biosensors

Over the last few decades, attempts have been made to utilize semiconductor field-effect transistors (FETs) in biological and chemical sensors due to their well-characterized behavior and ease of use of portable devices. In addition, the electrical measurement of bimolecular interactions is particularly desirable due to its adequacy for low-cost mobile sensors that could be used in the area by non-technical individuals [110]. FET-based biosensors are excellent candidates for several label-free transducers. They have gained much more attention in recent years for their considerable advantages, such as high scalability, ultra-sensitivity, rapid real-time analysis, inherent amplification, reduced power needs, direct electrical read-out, and low-cost bulk development, as opposed to surface plasmon resonance, microcantilever sensors, fluorescence instruments, and other approaches [12,111]. 

An FET-based biosensor includes three electrodes—the source, the drain, and the gate—such that the section between the drain and the source functions as a biological detection component that interacts with the target analyte/biomolecules and senses their presence, concentration, and electrical activity. The biosensor then converts the biological information directly into a detectable signal [112]. Subsequently, based on the implementation, the signal acquired can be illustrated, amplified, stored, and analyzed or sent to the data center for additional processes (Figure 6) [113]. The function of the FET-based biosensor can be encapsulated as: (1) an alteration in the concentration of the sample solution leads to a difference in the charge close to the interface of the sensor; (2) this shift of charge causes a decrease in the effective gate voltage; (3) this variation in the effective gate voltage contributes to a change in the current flow of the drain [114].

Due to the linear dispersion relation of the charge carriers in graphene, they have a very small effective mass and excellent carrier mobilities [115]. Furthermore, graphene-based nanomaterials have high thermal conductivity, as well as availability and low cost. The charge transfer nature of graphene enables a special channel material that replaces silicon and other common semiconductors. In comparison to other semiconductors, graphene does not require impurity doping to conduct electricity. It demonstrates self-doping phenomena that cause the carrier type and concentration to be regulated with the aid of an external electrical field. The alternating ability between GFET’s n- and p-channel varies from other FET technologies [116,117,118].

As a conductive channel substrate, many findings have been obtained through research on the contact resistance between graphene and metal to enhance the efficiency of the field-effect transistor. In addition, the need for high carrier mobility and luxurious functional groups has prompted a great deal of effort, such as the heating-assisted spray system for manufacturing mass production of RGO sheets and the interlayered quantum dots for elevating carrier mobility. Biocompatibility, ease of functionalization, and biocatalysis in the oxygen reduction reaction (ORR) are other outstanding features of graphene-based nanomaterials that are useful in the fabrication of GFETs for ultrasensitive and low-noise cancer detection [118,119,120].

It is important to maintain the stability of the antibody-modified surface during FET measurements. However, unlike other measurement methods, FET measurements use an electrical application on the gate area that can damage the surface. Electrical application may repel antibodies that are immobilized on the surface. Thus, preserving antibody-modified FET properties during measurements improves the reliability of FET antigen detection. Hidshima et al. evaluated three types of FET-modified antibodies to measure stability. The change in FET properties was determined three times in a row by calculating the change in the amount of threshold voltage change (C_v_). Modified C_v_ FET with electrically activated antibodies showed good stability [121].

In 2017, Zhou et al. developed and characterized a label-free antibody-modified graphene immunosensor based on a noncovalent modification to detect carcinoembryonic antigen (CEA). In this study, anti-CEA was immobilized in a single-layer graphene channel using a PYR-NHS linker. The anti-CEA-modified GFET could achieve a limit of detection (LOD) of CEA less than 100 pg·mL^−1^ in real-time, with the minimal nonspecific binding of non-target proteins [122]. In other research, Chen et al. used a polypeptide functionalized RGO-FET biosensor to accurately detect the matrilysin (MMP-7) in clinically relevant concentrations. This biomarker specifically digests negatively charged JR2EC immobilized on RGO, thereby modulating the conductance of RGO-FET. The proposed assay has demonstrated the detection of MMP-7 in human plasma with an LOD of 40 ng·mL^−1^ [123].

In 2013, Kim et al. applied the R-GO FFET for label-free ultrasensitive diagnosis of a prostate cancer biomarker of α1-antichymotrypsin (PSA-ACT). When PSA-ACT complexes were placed on the R-GO channel substrate using monoclonal PSA, it caused a linear reaction of the gate voltage (ΔVg, min), caused by the protein–protein interactions on the R-GO FET sensor. They reported this detection of protein–protein interactions up to the femtomolar level with a dynamic range over six orders in the Vg, using minutes of the alteration as a sensitivity parameter. Finally, the R-GO FET biosensor successfully demonstrated high specificity in the diagnosis of SA-ACT of the femtomolar level [124].

To increase the electrochemical reactivity of the FET biosensor system, Majd et al. designed an ultrasensitive aptasensor system based on multi-walled carbon nanotubes and RGO for label-free detection of ovarian cancer antigen 125 (CA125). The sensor has a simple design based on the noncovalent bonding of MWCNTs/conjugated aptamers on multilayer Graphene Oxide nanosheets and its integration with polymethyl methacrylate (PMMA) as a suitable substrate for designing flexible field-effect transistors. The presence of MWCNTs caused the immobilization of molecules due to the high surface-to-volume ratio. The designed aptasensor for CA125 (1.0 × 10^−9^–1.0 U/mL) demonstrated a wide linear dynamic range with a low detection limit of 5.0 × 10^−10^ U/mL. During the process, the proposed FET currents are present in the presence of various interfering species commonly found in biological fluids, including CEA (1.0 × 10^−3^ U/mL), AFP (1.0 × 10^−3^ U/mL), and CA15-3 (1.0 × 10^−3^ U/mL). Comparison with CA125 (1.0 × 10^−5^ U/mL) was measured. According to the values measured in the bar graph of Figure 7, the flow reduction is relative to the void (0.0 U/mL CA125) for CEA (1.4%), AFP (1.5%), and CA15-3 (3.2%), significantly smaller than that produced by CA125 (40.4%). These results confirm that the response of the RGO FET biosensor is induced by the binding of CA125 marker tumor to CA125 aptamer. In addition, the aptasensor successfully passed the CA125 detection test in real human serum samples [125].

### 5.2. Graphene-Based Surface Plasmon Resonance (SPR) Biosensors

SPR-based technologies have proven to be one of the most effective tools for real-time tracking of molecular dynamics, alongside quantitative measurement of numerous biomarkers such as proteins, DNA, entire cells, etc. [126,127,128]. One of the most critical aspects of the SPR biosensors is the surface of the sensor, as it plays a crucial role in the overall efficiency of the sensor. Many experiments have also concentrated on the use of smart sensing layers for more adjustable implementations [129,130,131]. SPR sensing is a powerful, label-free method for investigating noncovalent molecular interactions as a non-invasive method, and in the last two decades, SPR has been widely used in the study of noncovalent protein–DNA, protein–cell, DNA–RNA, DNA–DNA, protein–protein, etc., interaction experiments [132].

SPR biosensors have the potential for extensive implementation in biomarker diagnostics due to the very high surface plasmons’ sensitivity to the alternation in the reflective index (RI) of the dielectric medium. Interaction between the immobilized receptors on the metal surface and the analyte molecules causes a variation in the sensed medium’s refractive index (dielectric), which leads to an alteration in the propagation constant of the surface plasmons. This phenomenon affects the resonance condition of surface plasmons with specific surface plasmon waves (SPW) that have interactions with the incident p-polarized light of the same propagation constant. The energy of light photon transfers to the surface plasmons at the resonance angle, and the reflectance of the light significantly decreasing, results in forming a sharp dip in the SPR curve (reflectance with respect to the incident angle) [133,134]. The measured interacting optical wave can be used to fabricate various types of SPR biosensors with angular, intensity, phase, or wavelength modulation [135]. The Kretschmann configuration is the most common type of SPR setting based on attenuated total reflection for surface plasmons’ excitation [136]. Principles of SPR biosensor shown in Figure 8.

Phase detection based on coherence is usually very sensitive to mechanical movements in optical components and ambient noise, which leads to phase measurement errors. The stability of the SPR biosensor in the time axis is also required to monitor the biological response over time.

Wang et al. evaluated an efficient SPR phase biosensor for targeted drug delivery screening. The sensitivity of the sensor was determined by the SPR signal of a set of concentrations of sodium chloride solution, and its stability was estimated by oscillating the SPR signal of the system during the 80 min measurement process with a sample containing 0.9% sodium chloride. A significant reduction in ideal stability noise of 6 × 10^−7^ RIU/RU was reported during the 80 min measurement, which is similar to several high-sensitivity-phase SPR systems [137].

Graphene-based nanomaterials have a very high surface-to-volume ratio, which enhances the biosensor sensitivity due to the efficient light absorbing on the sensing layer by p–p stacking interaction between biomolecules’ carbon-based ring structure hexagonal units of graphene sheets [138]. Other superiorities such as high optical transparency, low reluctance, high carrier mobility, and tunability are also very promising for the fabrication of highly sensitive SPR biosensors [139,140,141,142]. It has been indicated both theoretically and experimentally that graphene-based nanomaterials are capable of effective excitement and propagation of surface plasmons [143,144]. Although the utilization of graphene-based nanomaterials can significantly enhance the sensitivity to the RI change of the SPR biosensor due to its extraordinary optical absorption and plasmonic characteristics, often the signal-to-noise ratio (SNR) and quality factor (QF) of the biosensor can adversely be reduced [145]. Multiple graphene layers will raise the full width at half maximum (FWHM) in the SPR curve by widening the reflectance dip and also reducing the depth of the dip, so the resolution of the biosensor decreases as well [140]. Therefore, synthesizing novel graphene-based nanomaterials that can eliminate these drawbacks and maintain or even augment their advantages are of high priority in the SPR biosensor fabrication.

Biplob Hossain et al. conferred numerical modeling of a graphene-coated fiber-optic surface plasmon resonance (SPR) biosensor for detection of genetic biomarkers involved in early breast cancer by means of DNA hybridization. The technique used in this sensor is based on the attenuated total reflection (ATR) method to detect the hybridization of deoxyribonucleic acid (DNA) with individual point mutations in the BRCA1 and BRCA2 genes. To detect breast cancer biomarkers (916delTT and 6174delT), SPR and SRF angles can be used, which are insignificant for uncoordinated DNA strands, while being a significant complement to DNA strands. As shown in Figure 9, due to the addition of the shDNA (pDNA) sequence to the assay medium, the RI increases due to the increase in the assay concentration and eventually shifts the θSPR and SPRF curves. Complete hybridization of 916delTT and 6174delT with pDNA can be understood by shifting the θSPR and SPRF curves to the right. In fact, RI changes due to the stabilization of different concentrated biomarkers of breast cancer DNA molecules (916delTT and 6174delT), which affects the SPRF and SPR curves. Changing the θSPR and SRF graphs to the right indicates the detection of 916delTT-specific mutations and the 6174delT interaction with pDNA, which also indicates the diagnosis of breast cancer. Additionally, numerical results showed that the use of graphene could be more sensitive compared with the usual SPR biosensors [139]. Furthermore, in another study, Habib et al. compared the performance of a graphene-MoS_2_ layered surface plasmon resonance (SPR) biosensor with a graphene-coated SPR biosensor. The numerical values obtained showed that the SPR biosensor coated with graphene-MoS_2_ compared to a single layer of graphene SPR biosensor is 175% more sensitive [146]. In other research, Wang et al. proposed a dual-channel fiber-optic SPR-LSPR biosensor. In this sandwich structure, the sensing channel was coated with a goat anti-human IgG-modified Au/GO bilayer to detect human IgG-labeled AuNPs. The presence of GO increased the loading of biomolecules, so the response signal was strengthened. The biosensor had an index sensitivity of 13,592 nm·RIU^−1^ with the LOD of 15 ng·mL^−1^, which was 15.3 times lower than the conventional SPR biosensor for detection of the same biomarker [147].

### 5.3. Graphene-Based Fluorescent Biosensors

Biomolecular imaging and biomarker detection can be performed using fluorescent nanomaterials and labels, a highly sensitive and selective method with adequate spatiotemporal resolution and low cost for application [148]. Numerous fluorimetric diagnostic tests and fluorescent-based biosensors focused on biocatalyst behavior have been documented using organic dyes, inorganic semiconductor quantum dots (QDs), and carbon nanomaterials as fluorimetric indicators [149,150]. The operation of these biosensors is based on the fluorescence phenomenon that happens when a fluorophore or fluorescently labeled molecule absorbs the corresponding electromagnetic radiation. According to the signal generating technique in fluorescent biosensors, they are categorized into four types, including FRET (Förster Resonance Energy Transfer), FLIM (Fluorescence Lifetime Imaging), FCS (Fluorescence Correlation Spectroscopy), and FI (change in fluorescence intensity) [151]. The basis of fluorescence-based detection is shown in Figure 10. The main advantages of fluorescence-based biosensors are extreme sensitivity, they are minimally invasive or non-invasive, they have the ability to utilize fluorescence intensity and fluorescence lifetime, and they provide the structure and microenvironment of molecules [152,153,154].

For removing the technical obstacles of conventional organic labels, many fluorescence-emitting nanomaterials such as dye-doped silica nanoparticles (DDSNs), fluorescent gold/silver metal nanoclusters, lanthanide-doped nanomaterials, upconversion nanoparticles (UCNPs), carbon nanomaterials, and fluorescent semiconductor quantum dots (QDs) have been utilized in the fabrication of fluorescent biosensors [130,156]. Graphene nanomaterials are favorable candidates for modification of fluorescent biosensors due to their high surface-to-volume ratio and excellent distance-dependent fluorescence quenching capability on the basis of FRET [149,157].

Even though the complete fluorescence mechanism of carbon nanomaterials is not understood, various types of them such as carbon nanotubes, GO, RGO, GQD, CQD, etc., have been used in fluorescent biosensors [158,159,160,161]. Some of the carbon nanomaterials, such as graphene and GO, are fluorescent quenchers because of their large electron plane on the nanosheet, which performs FRET to quench fluorescence [162,163,164]. On the other hand, multicolor GQD, CQD, and CNT offer robust fluorescence that makes them very practical in optical biosensors [82]. The emission range of CNT is near-infrared, but the main demerit is that the cytotoxicity of them is not thoroughly studied [82]. A significant fluorescent feature of some CQDs and GQDs is their excitation-dependent emission; when the excitation shifts from ultraviolet to near-infrared, GQD and CQD emit in the correspondence wavelength [165,166,167,168], but many of them have a consistent emission peak even with moving the excitation wavelength [169,170,171].

In 2015, Zhang et al. showed that with a nanofiber membrane with electrospinning of Graphene Quantum Dots for dual-purpose fluorescent and electrochemical biosensors, the fluorescent intensity is almost uniform, which determines the excellent stability of PVA/GQD nanofiber membranes [172].

Xu et al. designed a fluorescent aptasensor to detect four biomarkers of AFP, HER2, CEA, and VEGF165 simultaneously in the living cell. GO sheets were modified by four fluorophore-labeled aptamers, including FAM, Cy5, and AF405, and after the formation of the aptamer–protein complex, these fluorophores detached from the GO sheet and biomarkers were detectable by four different colors. This procedure proved to be rapid, biocompatible, and highly specific for the diagnosis of related cancers [173]. In other research, Wang et al. fabricated a fluorescent nanobiosensor for detection of MUC1, CEA, and CA125 by using Ag/AuNCs and two types of AgNCs and related aptamers as biorecognition elements. Based on the designed biosensor, the standard treatment for MUC1 was established in the presence of different concentrations (1.33 ng/mL, 2 ng/mL, 2.67 ng/mL, 13.3 ng/mL, 20 ng/mL, 26.7 ng/mL,133 ng/mL, 200 ng/mL), and as shown in Figure 11, the fluorescence intensity of AgNCs-DNA1 gradually increases with increasing concentration of MUC1. Thus, increasing F-F0 versus MUC1 showed a linear range of 1.33–200 ng/mL with a correlation coefficient of 0.991 and a detection limit of 0.18 ng/mL in the signal-to-noise ratio of 3. As a result, it shows the appropriate and high sensitivity of this medium for measuring MUC1. The standard curves for CEA and CA 125 showed similar results. In this study, GO played the role of the quencher and caused an increase in the fluorescent signal of real samples with high sensitivity [174]. Wang et al. used the DNase I fluorescence amplification method based on the reaction between Graphene Oxide (GO) and DNA aptamer to detect colorectal cancer exosomes. They combined fluorescent-dye-conjugated aptamers, Graphene Oxide, and DNase I enzymes in serum samples. Graphene Oxide quenched the fluorescence of aptamers and fluorescence recovery after incubation with samples containing the CRC exosome. DNA aptamers were placed on the surface of exosomes by the DNase I enzyme, thus increasing the interaction between the fluorescent aptamer probe and amplifying the signal [175].

In 2021, MA et al. reported that they used DNA nanomaterial attached to the GO surface to detect liver tumors. To analyze the selectivity of target and non-target cells, these cells were stained with and without GO by nanomaterial DNA. In the absence of GO, the comparison between target and non-target cells was insufficient. However, in the presence of GO, diffusion in non-target cells was very weak compared to target cells, indicating the role of GO in the detection of target cells. In in vivo studies, the liver tumor was stained well with the GO-based DNA nanomaterial. The studied system showed that it is highly able to target liver cancer cells. It also works well in the detection of fluorescence imaging of liver tumors and chemotherapy [176].

### 5.4. Graphene-Based Electrochemical Biosensors

Electrochemical sensing essentially needs a reference electrode, a counter (auxiliary electrode), and a working electrode, sometimes termed as the sensing or redox electrode(Figure 12). The reference electrode, usually produced from Ag/AgCl, is held at length from the reaction site to preserve a specified and consistent potential [177]. The working electrode acts as a transduction component in the biochemical reaction, while the counter electrode links the electrolytic solution so that the current can be delivered to the working electrode. These electrodes ought to be conductive and have chemical stability [178]. According to the Institute of Clinical and Laboratory Standards (CLSI; EP05-A3, EP24-A2, EP25-A), a change factor (CV) of less than 10% is required for reproducibility, accuracy, and stability. Therefore, further improvements in biosensors, especially in the case of electrodes and intermediates, are two components that determine the reproducibility, accuracy, and stability of all electrochemical biosensors [179]. Platinum, gold, carbon (e.g., graphite), and silicone derivatives are thus widely utilized based on the analyte [178]. 

A wide range of molecular identification elements can be used by electrochemical biosensors to detect biomarkers. A number of advantageous electrochemical strategies for detecting cancer biomarkers include techniques such as voltammetry (CV) [48]. The use of two or three electrodes with a potentiostat in voltammetric methods makes it possible to apply the potential and thus measure the current. In this method, designing the surface structure of the biosensor is very important to identify the analyte, amplify specific interactions, and suppress nonspecific interactions. In recent years, the use of nanomaterials and electroactive complexes in bioassays by electrochemical methods has provided high sensitivity for biomarkers. These nanomaterials accelerate signal transmission by creating a synergistic effect between catalytic activity, conductivity, and biocompatibility. They prevent the effects of classical biosensors such as bio-enzymatic sensors and critical microenvironmental factors [7]. The pairing of different bioreceptors with nanomaterials, the widespread levels of these nanomaterials, and the simplicity of electrochemical detection techniques are some of the factors of the high compatibility of electrochemical biosensors based on nanomaterials [181]. From a vast range of graphene-based nanomaterials, graphene metal nanoparticles, nanocomposites, metal alloy nanoparticles, magnetic nanoparticles, nanowires, nanofibers, nanorods, carbon nanotubes, and carbon nanofibers are more used. The presence of graphene in these nanomaterials has some advantages, such as fast electron transportation, high thermal conductivity, good biocompatibility, and excellent mechanical flexibility, which enhances the electrochemical biosensors’ function [182]. In electrochemical sensing, different electroactive bioreceptors can be used with high sensitivity, and due to unique oxidized/reduced potential for each molecule, they also offer high selectivity. Furthermore, graphene nanomaterials have a low residual current, wide potential window, and easy surface renewability, and because of their large overpotential for O_2_ and H_2_ density on the edge-plane defect spots, they bring about a great deal of electron transfer for biospecies [51].

Modification of electrodes with graphene-based nanomaterials also causes electrochemical species to distribute on the surface uniformly, and their 2D structure provides a high surface-to-volume ratio, which results in a suitable substrate for detecting adsorbed analytes. The electron transfer rate directly correlated with the exponential distance between the electrode surface and electrochemically active center of the bioreceptor, and graphene-based modification can dramatically reduce this distance to facilitate the electron exchange. This phenomena takes place on the edge of graphene sheets and/or at defects in the basal plane by nanowiring between active electrochemical sites and electrode surfaces to transfer electrons directly that promote sensing applications [183,184].

Akbari et al. designed an electrochemical biosensor for prostate cancer biomarker detection using GO/Au nanostructures, and this article showed that the GO/AuNP composite improves the electrical conductivity and biocompatibility [185]. Heydari et al. have successfully designed a new electrochemical biosensor using an aptamer for early detection of prostate-specific antigen (PSA) in human serum by using the RGO-MWCNT/AuNP nanocomposite, and because of the presence of the RGO, the conductivity between the nanocomposite improves, and LOD was equal to 1 pg·mL^−1^ [186]. A DNA biosensor based on AuNP-modified GO for the diagnosis of breast cancer biomarkers for early diagnosis was designed by Ayman et al. In this work, the electrochemical signal enhancement was achieved via GO, and the presence of GO increased the surface area [187].

In 2020, Asadi et al. developed a graphene-modified glassy carbon electrode electrochemical biosensor to detect miRNA-21, a biomarker in early prostate cancer. A molecular linkage agent was used to immobilize DNA on the surface of a graphene-modified glass carbon electrode. Charge transfer resistance (Rct) was measured by electrochemical impedance spectroscopy before and after hybridization. The electrochemical biosensor demonstrated a linear impedimetric response between ΔRct and logarithm of miRNA-21 concentration ranging from 10^−14^ to 10^−8^ M with a correlation coefficient of 0.972 and a detection limit of 3 fM. The results show that the electrochemical biosensors of the graphene-modified glass electrode can be used as an alternative to conventional methods for detecting the early stages of cancer [188].

In 2021, Jozghorbani et al. investigated the diagnosis of carcinoembryonic antigen (CEA) using a label-free electrochemical immunosensor to tether the antibody to the electrode surface. In this study, CV and EIS techniques were used. They first coated the GC electrode surface with RGO. The CV measurements were performed in a 0.1 M solution of PBS (pH = 7.4) containing a 5 mM ferro-ferricyanide probe against the Ag/AgCl reference electrode at a scan speed of 25 mV s^−1^ and in the range of +0.8 to −0.4 V. The redox peak currents shown in Figure 13 were reduced by modifying the GC electrode with RGO. In addition, the reduction of peak currents after incubation of the RGO/GC electrode with anti-CEA antibody indicates that some of the electron transfer pathways are blocked, and thus the charge transfer resistance is increased by the amidation reaction of the limited antibodies to the Graphene Oxide functional groups. Finally, by immobilizing the CEA antigen on the surface of the anti-CEA/RGO/GC immunosensor, the peak current was reduced again, which could be due to the formation of a more insulating layer that prevents the redox-active species from spreading to the electrode. Additionally, electrochemical impedance spectroscopy was used to obtain a signal to determine the antigen concentration. Finally, the CEA was immobilized on the surface of the anti-CEA/RGO/GC immunosensor. EIS results, like the results of CV, showed that the current was attenuated, which could be due to the block of the electron transfer pathways and the increase in the charge transfer resistance (Rct). These results can be used as a good signal to determine the antigen concentration [189].

### 5.5. Graphene-Based Surface-Enhanced Raman Scattering (SERS) Biosensors

In Raman scattering, photons inelastically lose (Stokes) or gain (anti-Stokes) energy because of molecular vibrational events and represent information about the molecular structure enabling in situ and real-time detection [190]. Surface-enhanced Raman scattering (SERS) is a subset of Raman scattering, a widely used sensing technique in which when the molecules are adsorbed on corrugated metal surfaces such as silver or gold nanoparticles, inelastic light scattering by molecules is greatly improved [191,192]. By way of plasmonic nanostructures, it provides a million-fold improvement, making the efficiency of detection down to the level of single molecules. Two different pathways, namely electromagnetic enhancement and chemical enhancement, accomplish SERS enhancement [193]. The design of the direct and indirect detection SERS-based biosensor shown in Figure 14.

In bioanalysis, SERS-based biosensors have the following advantages: (1) ability to represent biomolecules’ intrinsic fingerprint molecular information and ultra-high sensitivity down to the single-molecule level; (2) narrow peak bandwidth relative to fluorescence spectroscopy with strong tolerance to photobleaching and photodegradation; (3) many ways to appeal to specific applications for signal enhancement substrates of varying sizes and shapes; (4) a greater depth of laser penetration enables both diagnosis and imaging in vitro and in vivo [194]. 

One key downside of traditional silver-based SERS substrates is their low physical stability due to oxidation, which has a strong effect on their sensitivity and quality of efficiency. SERS-active metallic nanostructures are typically protected by a stable protective coating or shield constructed from inert materials, such as metal oxides and carbon materials, to overcome this restriction [195,196]. Among them, graphene and graphene-based derivatives, such as GO and RGO, are becoming favorable due to their stronger SERS effects. The SERS signal enhancement observed is due to the contribution of a process of chemical enhancement arising from the results of charge transfer between the graphene substrate and the adsorbed molecules [191]. GO exhibits several distinctive Raman scattering features such as the high-frequency D (disordered) and tangential mode (G-band), which are sharp, obvious, and can easily be distinguished from fluorescence backgrounds, making it a good Raman signal reporter [197,198]. However, the Raman signal of Graphene Oxide is weak and can be scarcely employed in Raman detection, but the Raman signal of Graphene Oxide could be greatly enhanced when Graphene Oxide abuts noble metal nanoparticles.

Practical applications of SERS nanotags require SERS signal stability when nanotags are exposed to the constant light of relatively high-power laser light. These SERS-related problems limit the quantitative analysis of reading signals in sensor applications and prevent the development of a reliable, quantitative SERS-based detection platform. Therefore, the protection and stabilization of SERS nanotags are key factors in the development of reliable diagnostic assays. Smolski et al. showed that following the formation of an encapsulating layer on Raman molecules at the surface of gold nanoparticles improves signal stability in the long-term exposure to intense laser light. Modified molecules have higher SERS intensities and therefore show their stability over time [199].

In addition, several intriguing properties, such as high optical transparency, high carrier mobility, chemical inertness, and biological compatibility, are often combined with graphene and its derivatives [196]. A variety of processes, such as coating, simultaneous assembly of graphene and nanoparticles, or deposition of noble metal nanostructures on top of a pre-deposited graphene layer, have demonstrated advances in SERS efficiency with the integration of graphene or its derivatives into SERS-active substrates. Sub-picomolar analyte detection capabilities have been enabled by structures with a monolayer of graphene sandwiched between beam-lithography-produced Ag nanostars and Au nanoparticles [191]. The simultaneous assembly of GO nanoplatelets with shape-controlled AgNPs (octahedra) was shown to produce hybrid materials that improved up to 3-fold the SERS signal [200]. One of the largest SERS enhancement ratios recorded to date for graphene–metal systems (1700-fold), which was almost 115 times and 14 times larger than that of graphene on Ag film and graphene on Ag nanoparticles, was shown to give a nanoparticle-film gap (NFG) system created with graphene acting as a sub-nanospacer between an Ag film and Ag nanoparticles [201].

Yi et al. developed an SERS-based biosensor to detect tumor cells. In this system, RGO was sandwiched between Ag and Au nanostructures. The Ag-RGO-Au system has a strong chemical mechanism due to the charge transfer between RGO and the target molecules. The SERS sensor based on Ag-RGO-Au was tested for sensitive detection of tumor cells in the presence of human tumor cells, liver cancer cells (BEL-7402), and normal human liver cells (202 liver cells). Tumor cells do not show a characteristic signal in the range of 1000–3200 cm-1 in the Raman spectrum on the Si slide. Placing labeled tumor cells on the Ag-RGO-Au substrate causes significant differences in the Raman spectrum that may indicate tumor cells. In addition to the changes in the width and shape of peaks D and G compared to the peaks for Ag-RGO-Au, the intensity of the peaks also increases. Especially for peak D, there is a clear shift in the 25 cm^−1^ wave number as well as spectral narrowing. The chemical interaction between RGO and tumor cells can be seen from the distortion at the characteristic peaks of graphene. In connection with the chemical mechanism of RGO, the Ag-RGO-Au system has also been used to identify tumor cells without the use of biomarkers. The results showed that free-labeled tumor cells and normal cells, due to interaction with RGO located on the Ag-RGO-Au substrate, could be characterized by Raman changes at specific peaks). The results are shown in Figure 15) [202].

In 2018, Liang et al. used a label-free graphene/gold nanopyramid-based SERS biosensor to detect colon cancer p53 −/− cells from p53 +/+ colon cancer cells. The hybrid SERS biosensor is capable of detecting p53 −/− cells from the p53 +/+ cells of three different cell states, live, dead, and burst, with an average 81% sensitivity and 97% specificity. These results show the potential of this SERS graphene hybrid for cancer detection [203].

### 5.6. Graphene-Based Electrochemiluminescent Biosensors

Electroluminescence or electrogenerated chemiluminescence (ECL) is the shared area of electrochemistry and chemiluminescence (CL). In this mechanism, by applying a potential on the electrode’s surface, the electrochemical energy is being converted to radiative energy [204,205,206]. This phenomenon can happen by utilizing species that undergo electron transfer reaction to form an excited state and, after that, produce light when molecules return to the ground state [207,208]. Therefore, ECL does not need external light sources, so problems of light scattering inherent in photoluminescence (PL) can be avoided [206]. In the ECL method, the electrochemical reactions take place through the interplay of the luminophore and a co-reactant molecule by applying only one single potential step. Two approaches to the co-reactant ECL mechanism are available. The first one is the oxidative reaction, in which a potent reducing radical agent is produced by oxidation of the co-reactant in a homogeneous follow-up reaction. This radical can reduce the oxidized luminophore, so the luminophore becomes excited and emits light. The second one is reductive oxidation, in which the reduction takes place straightforwardly [209].

Due to the convergence of electrochemical and spectroscopic approaches, ECL has many benefits over chemiluminescence and photoluminescence, such as improved temporal and spatial modulation of light emission, high sensitivity due to lack of light excitation and a wide dynamic range, and fast detection in a low volume of sample. Furthermore, ECL has been successfully used for diagnostic purposes in real samples and complex matrices such as blood, urine, and cell lysates because of its remarkable signal-to-noise ratio (S/N) [209].

Nanoparticles possess outstanding electrochemical, photonic, and magnetic properties that make it possible for ECL sensing designs to be superior transducers [210,211]. Due to influences such as its high quantum Hall effect and electron–hole symmetry, graphene and its derivatives exhibit extraordinary electrical conductivity [212,213]. In order to enhance electrical efficiency, the two-dimensional RGO can be modified, and overall, the nanobiosensor displayed strong magnetism, ECL properties and biocompatibility, and low toxicity. In addition, in terms of linear range, stability, reproducibility, selectivity, and sensitivity, the high analytical output is demonstrated by the nanobiosensor [214]. GO-based ECL aptasensing is shown in Figure 16.

Although the exact PL process of CQDs and GQDs is still argumentative, π-plasmon and surface or edge defects are related to the broad optical absorption and photoluminescence emissions of carbon-based QDs. The few PL colors and poor quantum yield are still the main shortcomings of CQDs and GQDs in sensing applications. In addition, surface passivation may also impact CQDs’ ECL behaviors. The oxidized CQDs have outstanding ECL behaviors without any surface passivation from a top-down technique. However, the CQDs passivated via a bottom-up route with some capping reagents, displaying erratic ECL signals with elevated background interference [215]. CDs were immobilized on graphene in the CDs/S_2_O_8_^2−^ system, which could promote both C^•−^ and SO_4_^•−^ production in the ECL system to produce more C^*+^, resulting in an almost 48-fold amplification of the ECL [216].

After reviewing the types of graphene-based biosensors, Table 2 compares the different techniques for detecting cancer biomarkers.

## 6. Challenges and Opportunities for Graphene-Based Biosensors

Recent advances in the development of biosensors were discussed in this study. Recent research on biomarkers over the past decade has shown that biosensors can detect cancer in the early stages with minimal amounts of biological material and various diagnostic methods. Despite the invention of various biosensors, all of these sensors still have advantages and disadvantages. Powerful portable and user-friendly biosensors still have challenges as long as they become a reliable diagnostic method for cancer diagnosis. Challenges and future prospects in this field are rapidly being explored and developed. The issue is, what has led to the rapid progress in this area? Most research refers to nanomaterials or nanoscale measurements, or in many cases both, as the cause of rapid progress. Research and developments of current nanobiosensors are focused on reducing the mentioned problems such as high sensitivity and selectivity, accuracy and precision, and cost reduction. Hence, various methods have been used to minimize the challenges in biosensors. The most important solution proposed is the integration of different mechanical, electrical, chemical, and biological systems using special nanomaterials. If this sequential process occurs, the new diagnostic nanobiosensors could be extended to highly sensitive cells for cancer diagnosis and treatment without serious side effects [99].

Limit of detection (LOD) is a critical performance characteristic that requires careful evaluation during method validation. LOD is generally expressed as the amount of analyte that the analytical method detects in at least 95% of cases. Cancer dosage markers may be very low, so the use of high-LOD methods puts the researcher at risk of negative diagnosis. Very low LODs allow the detection of target biomarkers without enzymatic amplification. Therefore, the development of new techniques with selectivity, sensitivity, and low limit of detection (LOD) is very important in routine diagnosis [257]. For this purpose, many alternative analytical methods, such as graphene-based biosensors with higher sensitivity, have replaced conventional methods such as PCR and ELISA. One of the most important roles of nanomaterials is to pre-concentrate the analyte and lead it to the detector. The high load of diagnostic species on the nanoparticles facilitates bonding. Nanoparticles also provide the ability to detect lower amounts of analytes. This function makes it possible to measure fluids with lower analyte concentrations, which is not possible with most existing sensors [258]. 

In this regard, Hussein et al. reported in 2019 that the limit of detection (LOD) of the biosensor was about 2 ng/mL, which is 50 or 60 times lower than the detection of antigen by ELISA treated with gold nanoparticles (LOD = 100 ng/mL) and conventional ELISA [259].

Circulation of tumor cells (CTCs), found in the peripheral blood of cancer patients, can be used to detect cancer early. With CTCs, 1 cell per 10 mL can be important, while for pathogens, it can be one microbial species per liter in water quality analysis. Larger volumes pose more challenges for biosensors because these low concentrations require multiples of the sample to be analyzed. Such large volumes can be challenging even for microfluidics, although the separation of CTC by Warkiani et al. solves the higher volume challenge [260]. 

In this regard, Pudineh et al. examined the heterogeneous phenotypes of circulating tumor cells (CTCs) in whole blood to discover the complex and dynamic features of these potentially vital clinical markers. This task is challenging because these cells are present at parts per billion levels among normal blood cells. To characterize CTCs, they developed a new method based on nanoparticle activation called cytometry. Accordingly, it indexes CTCs based on their superficial expression phenotype. They used a microfluidic chip to process whole blood samples [261]. It is this ability to measure very small amounts of analyte that guarantees a bright future for measurement research. 

In addition, to solve the mass transfer problem using dispersed electrodes for PSA detection, Wu et al. compared the performance of finite dispersible electrodes with a similar sensor prepared in the same way on a normal flat electrode. The dispersible electrodes were used to solve the problem of mass transportation with very low detection sensors. This features the detection limit from the flat sensor to the dispersible electrodes reduced from 7.05 pM to 3.0 fM. Similarly, detection limits of up to 15fM or even lower have been reported for a signal increase [258].

Ideally, for graphene-based biosensors, we need sensors with high response to the target analyte or molecule that have little or no response interference species. To achieve this goal, more working probes must be used to improve the detection of the target analyte or molecule [262]. Despite recent advances, nanomaterial-based electrochemical bioassay still faces serious challenges, including:Stability of nanobiosensors in electrodes.High sensitivity of biosensors to changes in environmental and medical conditions.Ability to reuse nanobiosensors.Problems caused by biocompatibility or non-toxicity to the environment.Complex technology for making electrochemical nanobiosensors.The complexity of the interaction method of nanomaterials and biomolecules.Processing, creating special features, and connection problems.Access to high-quality nanomaterials and the nature of these nanoscale compounds on the electrode plate [99].

Reproducibility, controllability, and scalability are other challenges that need to be overcome. Detecting cancer biomarkers in real samples such as blood plasma, etc., is also a challenge. Because real samples have different proteins, ions, and chemical species, they can be misdiagnosed. One possible way to solve this problem is to disable the sensor surface with an antifouling agent to focus on the target [263]. Thus, as discussed in the properties of nanomaterials, given the ability of graphene-based biosensors to combine their outstanding chemical and physical properties, it seems to be a turning point in the long road to achieving early and efficient cancer diagnosis. Despite great progress, there are still challenges to overcome. In the near future, it is hoped that graphene-based devices will be developed that can detect multiple cancer biomarkers simultaneously. 

## 7. Conclusions

Today, biosensors are used as a sensitive and rapid diagnostic tool in the early detection of cancers. In addition to making early diagnosis and treatment options more accessible to patients and improving their quality of life, biosensors can develop multiple sensors, differential and semi-selective, that can measure multiple markers in a single unit. The biosensors are able to read direct responses as positive and negative responses of biological fluid, including biomarkers, and also lack limitations such as throughput limitations and small sample volumes. Because circulating cancer biomarkers are generally found in small amounts in the background of non-target cells, EVs, and biomolecules, the combination of magnetic nanomaterials and electrochemical biosensors is appropriate to overcome this diagnostic challenge. The use of nanomaterials in the manufacture of biosensors for the detection of biomarkers is significant because these cases increase the range of detection and increase the sensitivity of the biosensor. Graphene and its derivatives (GO, RGO, GQDs) are used in the fabrication of biosensors because of their high surface area, optical properties, and high thermal and electrical conductivity. In this study, the methods of synthesis of graphene and its derivatives were investigated. Oxidative exfoliation-reduction, CVD, and LPE have a high potential for industrial implementation. Finally, in order to synthesize graphene, the environmental hazards, cost, and quality of synthesized graphene should be considered. The use of graphene-based materials for biosensing has achieved great success in a short time. Recent studies in this literature review indicate that, among various detection methods, electrochemical graphene-based cancer biosensors are the best option, due to their high sensitivity in a rapid assay. Graphene-based biosensors show high sensitivity and can be employed for the detection of cancer, and graphene is a super new material for biosensing. The future development direction of graphene-based biosensors should be more portable, reproducible, miniaturized, and high-throughput in detection, and the future development of these 2D graphene nanomaterials will be further developed and tailored for the specificity of receptors.

## Figures and Tables

**Figure 1 biosensors-12-00269-f001:**
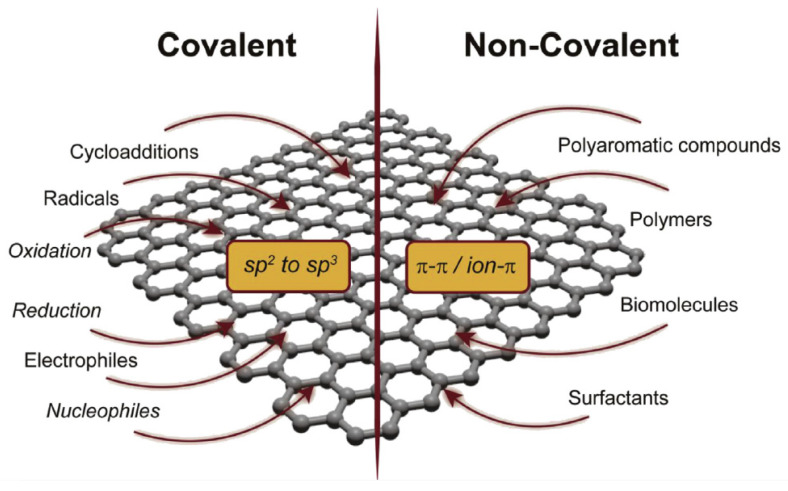
General representation of functionalization possibilities for graphene nanomaterials. Reprinted with permission from ref. [54]. Copyright 2019, Elsevier.

**Figure 2 biosensors-12-00269-f002:**
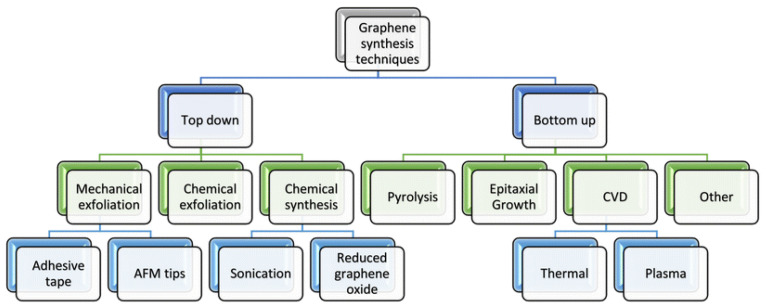
A process of graphene synthesis methods. Reprinted with permission from ref. [66]. Copyright 2016, Springer Nature.

**Figure 3 biosensors-12-00269-f003:**
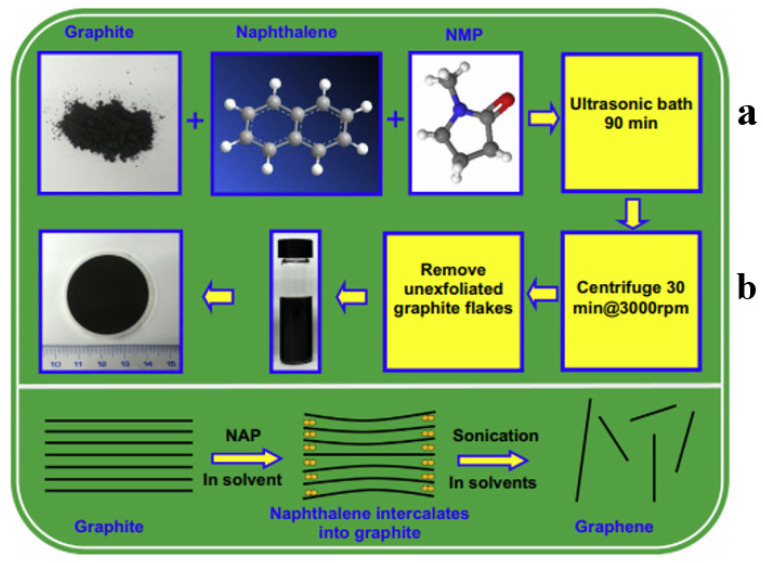
(**a**) Schematic of the exfoliation of graphene from graphite powder with the excess of naphthalene. The procedure includes different operations, including mixing, exfoliation, centrifugation, and vacuum filtration. (**b**) The illustration of the exfoliation of graphite using the π–π stacking interaction with naphthalene. Reprinted with permission from ref. [72]. Copyright 2014, Elsevier.

**Figure 4 biosensors-12-00269-f004:**
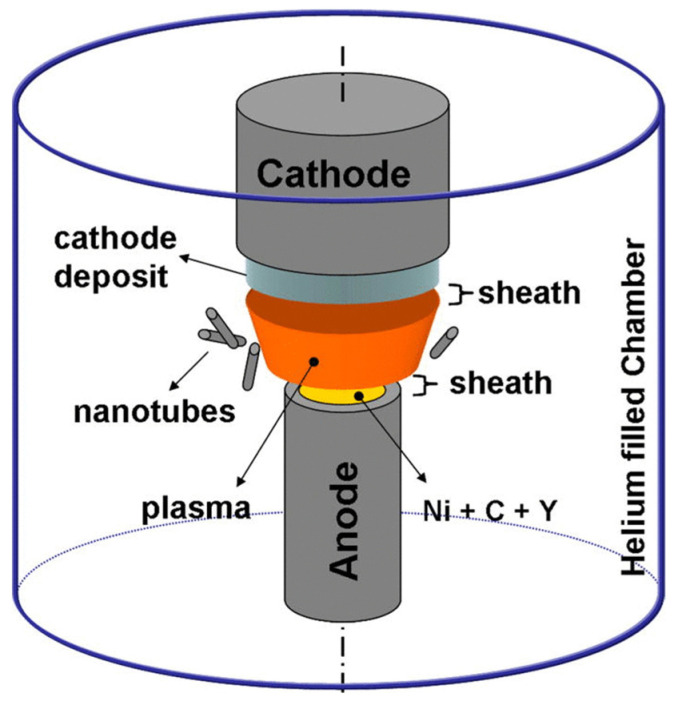
Schematic of the setup of the arc discharge system. Reprinted with permission from ref. [76]. Copyright 2012, AIP Publishing.

**Figure 6 biosensors-12-00269-f006:**
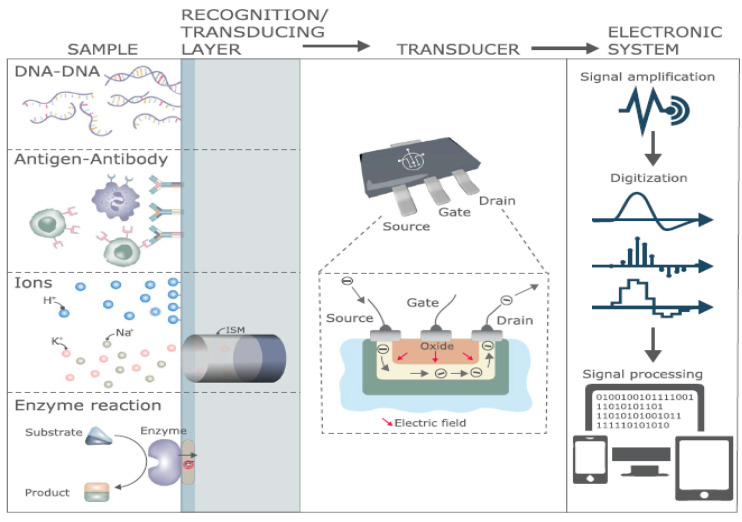
Depiction of a chemical and biological FET sensor. Reprinted with permission from ref. [113]. Copyright 2017, Elsevier.

**Figure 7 biosensors-12-00269-f007:**
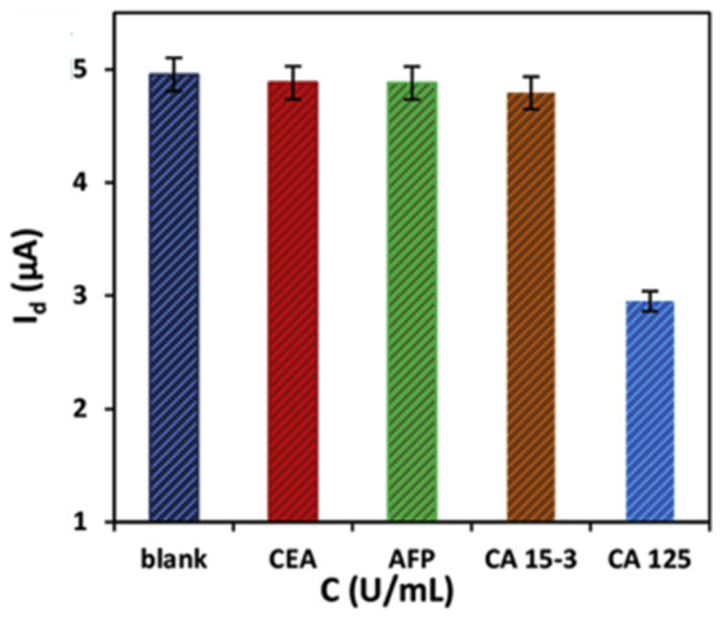
A histogram details the sensing performance of the few-layer RGO FET aptasensor. Reprinted with permission from ref. [125]. Copyright 2018, Elsevier.

**Figure 8 biosensors-12-00269-f008:**
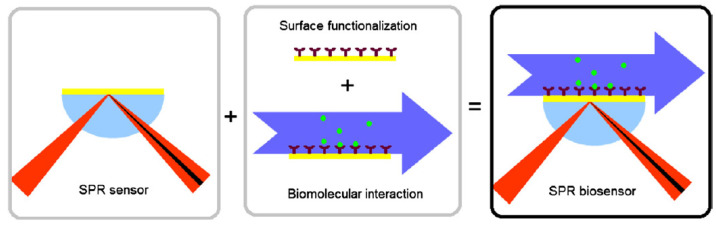
Principles of SPR biosensor illustration. Reprinted with permission from ref. [135]. Copyright 2011, Elsevier.

**Figure 9 biosensors-12-00269-f009:**
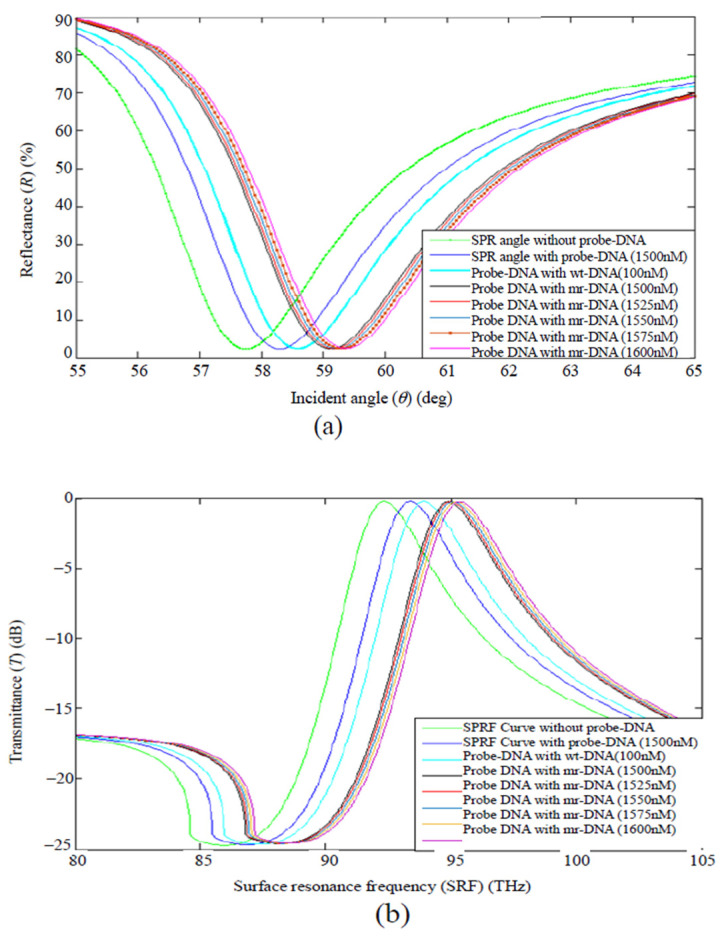
Change in characteristics curves for (**a**) SPR curve characteristics and (**b**) SPRF curve characteristics of the SPR sensor without pDNA, with pDNA, and with flowing different concentrated complementary target mrDNAs or mismatched target wtDNAs of BRCA1 and BRCA2 genes with PBS solutions. Reprinted with permission from ref. [139]. Copyright 2019, Springer Nature.

**Figure 10 biosensors-12-00269-f010:**
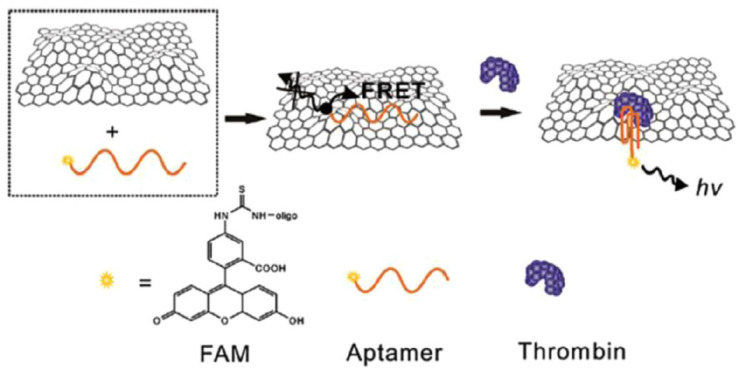
Schematic representation of fluorescence-based detection. Reproduced from Reprinted with permission from ref. [155]. Copyright 2017, Elsevier.

**Figure 11 biosensors-12-00269-f011:**
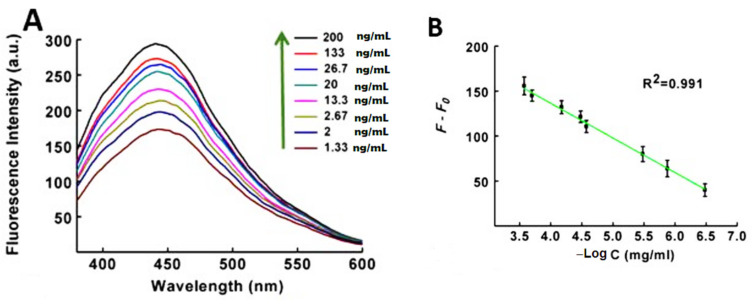
(**A**) Fluorescence spectra of the assaying model in the presence of varying concentrations of MUC1. (**B**) The linear relationships between F-F0 and the concentrations of MUC1. Reprinted with permission from ref. [174]. Copyright 2018, Elsevier.

**Figure 12 biosensors-12-00269-f012:**
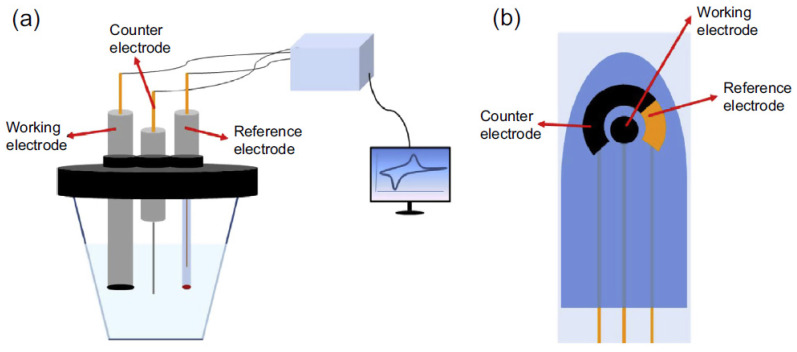
(**a**) An electrochemical working cell, (**b**) disposable screen-printed electrode. Reprinted with permission from ref. [180]. Copyright 2017, Elsevier.

**Figure 13 biosensors-12-00269-f013:**
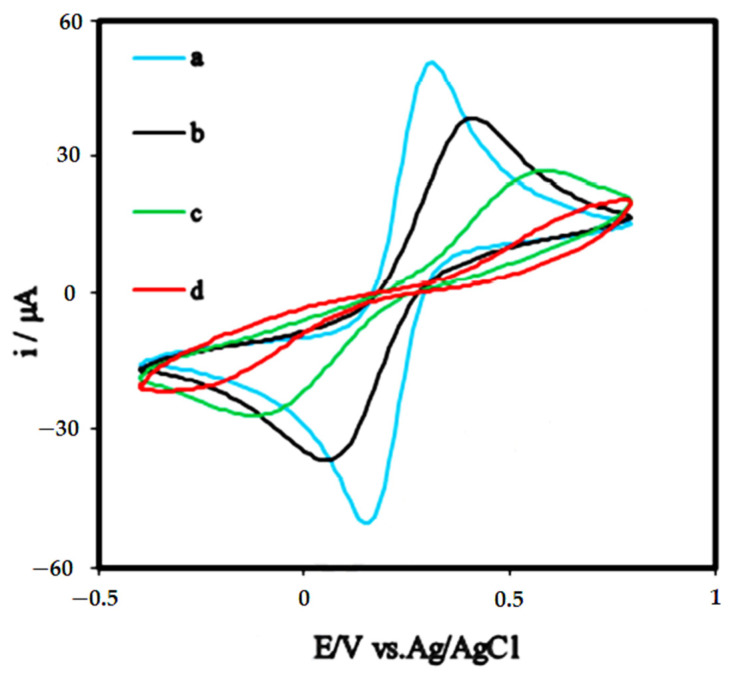
Cyclic voltammogram data recorded for (**a**) GCE, (**b**) RGO/GCE, (**c**) anti-CEA/RGO/GCE, and (**d**) anti-CEA/RGO/GCE biosensor incubated with CEA protein at a scan rate of 25 mV s^−1^ in a 0.1 M PBS solution (pH 7.4) containing 5 mM of the ferro-ferricyanide probe. Reprinted with permission from ref. [189]. Copyright 2021, Elsevier.

**Figure 14 biosensors-12-00269-f014:**
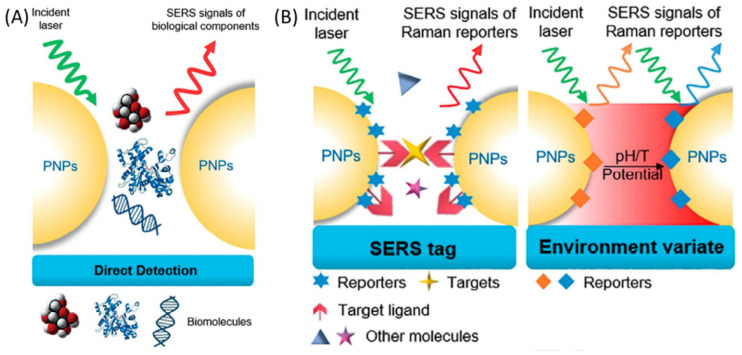
Illustration of the design of the SERS-based biosensor. (**A**) Direct SERS detection in which the intrinsic SERS signals emit from the substrate without labeling molecules; (**B**) indirect SERS detection in which the generated SERS signals from Raman reporters alter due to the presence of bound targets or change in the environmental properties. Reprinted with permission from ref. [190]. Copyright 2020, Elsevier.

**Figure 15 biosensors-12-00269-f015:**
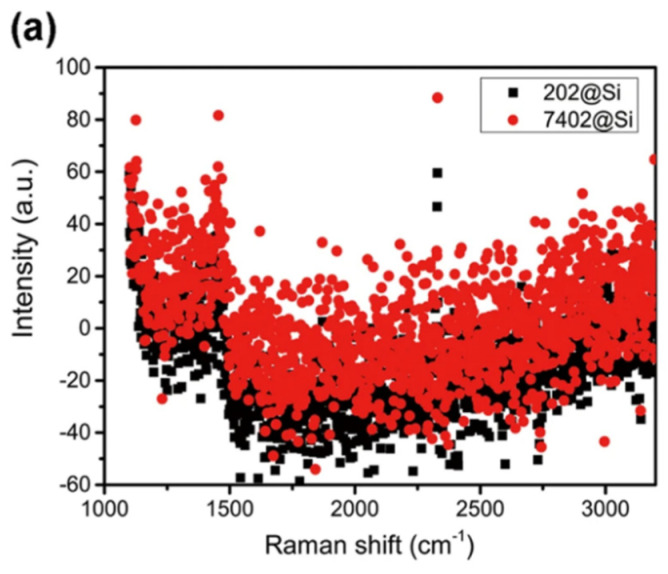
Identification and detection of tumor cells via SERS of Ag-RGO-Au. (**a**) Conventional Raman spectra of tumor cells (7402, red circle) and normal live cells (202, black square) on the silicon slide, which are indistinguishable in the vibration information. (**b**) Raman spectra of RGO on Si slide, RGO on Au nanostructures, GFP labeling tumor cells on Si slide, GFP labeling tumor cells on G-SERS substrate and G-SERS substrate. (**c**) Raman spectra of the same tumor cells (7402, red dash line), normal live cells (202, blue solid line) in (**a**) on our G-SERS substrate (green dash dot line). Reprinted with permission from ref. [202].Copyright 2016, Springer Nature.

**Figure 16 biosensors-12-00269-f016:**
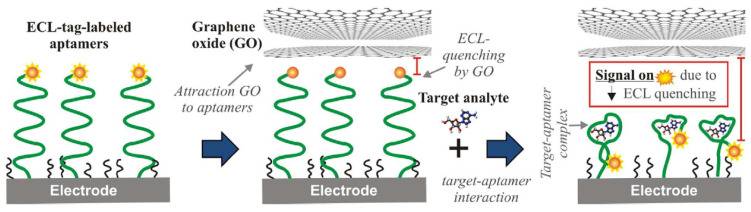
Illustration of GO-based ECL aptasensing. Reprinted with permission from ref. [215]. Copyright 2017, Elsevier.

**Table 2 biosensors-12-00269-t002:** Comparison of different cancer biomarkers detection techniques.

Method	Interface	Biomarker	LOD	Dynamic Range	Type of Cancer	Ref.
FET	RGO/streptavidin (SA)	Biotinylated microvesicles (B-MV)	20 particles·µL^−1^	10^5^ to 10^6^ particles·mL^−1^	Various cancers	[217]
FET	pSF/GO-TAPP/RGO/aptamer (AS1411)	MDAMB-231	-	10 to 10^6^ cells·mL^−1^	Breast cancer	[218]
FET	GOSS/pentacene/HER2 antibody	SkBr3 cells	100 cells·μL^−1^	-	Breast cancer	[219]
FET	G/PBASE/anti-AFP	α-fetoprotein (AFP)	12.6 ng·mL^−1^	44.9 to 784.9 ng·mL^−1^	Hepatocellular carcinoma	[220]
FET	G/MWCNT/aptamer	CA125	0.5 nU·mL^−1^	10^−9^ to 1 U·mL^−1^	Ovarian cancer	[125]
FET	Ab-MGLA/poly-SiNW	APOA2 protein	6.7 pg·mL^−1^	19.5 pg·mL^−1^ to 1.95 μg·mL^−1^	Bladder cancer	[221]
FET	RGO/Ab	PSA-ACT	100 fg·mL^−1^	10^2^ to 10^9^ fg·mL^−1^	Prostate cancer	[124]
SPR	GO-COOH	CA199	10 unit·mL^−1^	-	Pancreatic cancer	[222]
SPR	G/FA	folic acid protein (FAP)	5 fM	5 to 500 fM	Prostate cancer	[223]
SPR	Au/Cys/GO/COOH/Ab	cytokeratin 19 (CK19)	1 fg·mL^−1^	0.001 to 100 pg·mL^−1^	Lung cancer	[103]
SPR	Au/Cys/Carboxyl-GO-Peptide	hCG protein	1.15 pM	1.15 to 28.7 pM	Choriocarcinoma	[224]
SPR	Cr/Au/cys/Carboxyl-GO-anti PAPPA2 protein	PAPPA2 protein	0.01 pg·mL^−1^	0.1 to 10,000 pg·mL^−1^	Choriocarcinoma	[225]
SPR	Anti-CEA pAb/Au-Anti Mouse IgG-PDA-RGO and POEGMA-co-GMA-anti-CEA mAb	CEA	500 pg·mL^−1^	0 to 16 ng·mL^−1^	Various cancers	[226]
SPR	Au/POEGMA-co-GMA-Mouse anti-AFP and Anti-Rabbit IgG-RGO/Ag/Rabbit anti-AFP	α-fetoprotein (AFP)	100 pg·mL^−1^	1 to 100 ng·mL^−1^	Hepatocellular carcinoma	[227]
SPR	Apt/AuNP/GO	ssDNA	0.2 fM	10^− 15^ to 10^−11^ M	Breast cancer	[228]
Fluorescence	GO/aptamer/FAM+Dnase I	MUC1	10 pg·mL^−1^	50 pg·mL^−1^ to 100 ng·mL^−1^	Breast cancer	[229]
Fluorescence	UCPs–aptamer–CNPs	CEA	-	0.1 to 40 ng·mL^−1^	Various cancers	[230]
Fluorescence	GQD-CuNC/aptamer	HTLV-I DNA	10 pM	20 pM to 12 nM	Leukemia	[231]
Fluorescence	HE4 antibody/red and green GQDs	HE4	4.8 pM	4.8 pM to 300 nM	Ovarian cancer	[232]
Fluorescence	CNSs/P0-FAM	MUC1	25 nM	0 to 6 μM	Breast cancer	[233]
Fluorescence	Apt/UCNP/GO	CEA	10.7 ng·mL^−1^	0.03 to 6 ng·mL^−1^	Various cancers	[234]
Fluorescence	GQD-PEG-aptamer/MoS_2_	EpCAM protein	450 pM	3 to 54 nM	Various cancers	[235]
Fluorescence	Apt/FAM/GO	PSA	0.76 pg·mL^−1^	1 to 100 pg·mL^−1^	Prostate cancer	[236]
Fluorescence	Apt/GelRed/GO	PSA	10 pg·mL^−1^	100 pg·mL^−1^ to 200 ng·mL^−1^	Prostate cancer	[237]
Fluorescence	Apt/FAM/GO	VEGF	0.256 nM	0.5 to 5 nM	Various cancers	[238]
Fluorescence	Apt/FAM/GO	VEGF	1 pM	5 to 200 pM	Various cancers	[239]
Fluorescence	Apt/FAM GO	AFP	0.909 pg·mL^−1^	1 to 150 pg·mL^−1^	Various cancers	[240]
Fluorescence	Apt/DNA GO	Exosomes	2.1 × 10^4^ particles·µL^−1^	-	Colorectal cancer	[139]
Electrochemistry	FA/GAM/OA	Liver cancer cells	5 cells·mL^−1^	5 to 10^5^ cells·mL^−1^	Hepatocellular carcinoma	[241]
Electrochemistry	FA/CuO/WO_3_-GO	AGS cancer cell	18 cells·mL^−1^	50 to 105 cells·mL^−1^	Gastric cancer	[242]
Electrochemistry	AuPd-ANPs/GQDs/ACF	Hydrogen peroxide	500 nM	1.0 μM to 18.44 mM	Breast cancer	[243]
Electrochemistry	MnO_2_/NWs/AuNPs/GF	Hydrogen peroxide	1.9 μM	0.01 to 9.51 mM	Breast cancer	[244]
Electrochemistry	SS-probe/GO/GNR	miRNA-155	0.6 fM	2.0 fM to 8.0 pM	Breast cancer	[245]
Electrochemistry	Mucin1 antibody- MB@GO-COOH-SPCE	Mucin1	0.04 U·mL^−1^	0.1 to 2 U·mL^−1^	Various cancers	[246]
Electrochemistry	SRGO-HD	8-OHdG	1 nM	20 to 0.002 μM	Various cancers	[247]
Electrochemistry	FA/Glu-GQD-Pd@Au	HepG2	2 cells·mL^−1^	3 to 105 cells·mL^−1^	Hepatocellular carcinoma	[248]
Electrochemistry	GO/AuNPs/Ab1 GO/AuNPs/Ab2	tPSA fPSA	0.2 ng·mL^−1^ 0.07 ng·mL^−1^	2 to 10 ng·mL^−1^ 0.1 to 2.2 ng·mL^−1^	Prostate cancer	[185]
Electrochemistry	Au/RGO/FA	folic acid protein (FAP)	1 pM	1–200 pM	Prostate cancer	[249]
Electrochemistry	Au/RGO	FA	1 pM	1 to 200 pM	Various cancers	[249]
Electrochemistry	Graphene/PBSE	miRNA-21	3 × 10^−15^ M	10^−14^ to 10^−8^ M	Prostate cancer	[188]
Electrochemistry	Au/GO	PSA	0.028 ng·mL^−1^ and 0.007 ng·mL^−1^	0.5 to 7 ng·mL^−1^	Prostate cancer	[109]
SERS	AgNPs/GO/Ab and biotinylated Ab/streptavidin-labeled Glucose oxidase	PSA	0.23 pg·mL^−1^	0.5 to 500 pg·mL^−1^	Prostate cancer	[104]
SERS	MWCNT/thionine-NH_2_-RGO−COOH-Ab	PSA	2.8 fg·mL^−1^	10 to 20 ng·mL^−1^	Prostate cancer	[250]
ECL	Anti-CEA/Au-FRGO-CeO_2_@TiO_2_	CEA	3.28 fg·mL^−1^	0.01 to 10 ng·mL^−1^	Various cancers	[251]
ECL	Anti-CEA/HM-GQDs-AuNPs	CEA	0.01 ng·mL^−1^	0.02 to 80 ng·mL^−1^	Various cancers	[252]
ECL	GCE/PPy-NH2GO-Ag_2_Se@CdSe-Ab/BSA	CA72-4	2.1 × 10^−5^ U·mL^−1^	10^−4^ to 20 U·mL^−1^	Gastric cancer	[253]
ECL	RGO/Au-CdS:Eu QDs/Ab	α-fetoprotein (AFP)	0.05 pg·mL^−1^	0.00005 to 1.0 ng·mL^−1^	Hepatocellular carcinoma	[254]
ECL	Au-CdS/capture DNA-PSA aptamer/Fc-G	PSA	0.00038 ng·mL^−1^	0.001 to 25 ng·mL^−1^	Prostate cancer	[255]
ELISA	PBS/hydrochloric acid/BSA	nivolumab	3.0 µg·mL^−1^	100 ng/mL–200 µg·mL	lung cancer	[256]

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
