# Peer review of "Properties and Applications of Graphene and Its Derivatives in Biosensors for Cancer Detection: A Comprehensive Review"

_biosensors, 2022, doi:10.3390/bios12050269_

Round 1
Reviewer 1 Report
Pourmadadi et al. Reviewed on the “Properties and applications of graphene and its derivatives in biosensors for cancer detection: A comprehensive review.” The content of the work is interesting, but the manuscript cannot be published in the present form due to the following issues:
- A lot of reviews are available of the usage of graphene as biosensors for cancer detection For Eg.:-https://link.springer.com/article/10.1007/s00604-020-4152-8, https://iopscience.iop.org/article/10.1149/2.0252003JES Therefore respond to the following quarries:- (a) What is the main question addressed by the research? (b) Do the author consider the topic original or relevant in the field, and if
so, why? - For clear understanding, a schematic should be added in the topic “Graphene-based materials synthesis methods” on Page 5 for both top-down and bottom-up methods.
- Authors should add one para suggesting why modification of graphene by different types of biorecognition elements is required and it should be added which form of graphene is very efficient functionalized or un-functionalized graphene
- Further, also add a chart in the section “Modification of graphene by different types of biorecognition elements” on page 7
- Please include the name of the characterizations related to the stability or reliability of the biosensors for the different applications.
- Table 1 does not include the latest references
- Grammar and lots of typological errors such as subscripts, superscripts, etc. are present in the present form of the manuscript. So need an extensive rectification
Author Response
- What is the main question addressed by the research?
The writers of this review investigated various methods of functionalizing graphene as covalent and non-covalent. On the other hand, we investigated various biorecognition elements that have been used in cancer diagnostics based on graphene. We also thoroughly investigated the many techniques of cancer detection with graphene, identifying the optimal methods of diagnostic and biorecognition elements.
- Do the author consider the topic original or relevant in the field, and if so, why?
The current review is an update on prior studies, and it includes additional references up to 2022. Furthermore, this is an original review in which several approaches to cancer detection based on graphene biosensors have been thoroughly investigated, whereas earlier studies have not done so. Some of the greatest biorecognition features that can be employed for cancer diagnosis are thoroughly addressed in this review, which has not been the case in past evaluations.
Other comments are covered and highlighted in the manuscript.

Reviewer 2 Report
The authors have prepared a comprehensive review about the development of graphene-based biosensors for cancer detection. The review is timely and interesting, however, there are some important revisions that must be done before a decision upon publication can be made.
- In order for the large and diverse audience of Biosensors, the authors must show some example data from each section of biosensors types. For example, what does the signal graph look like in FET-based biosensors? how do the researchers make the interpretation of that data and confirm the cancer detection? What does an SPR sensorgram look like? The plot is drawn using the changes in resonance units versus time, so how do the researchers make the cancer detection using the sensorgram data? The same type of information should be given for electrochemical, SERS and fluorescence examples. The authors can show the results from the papers that they have highlighted or shown the figures from. The authors have just shown the basic detection principles of these techniques but they did not give any examples how the data obtained from these techniques can be used for cancer detection.
- In scheme 1, the authors should correct the typo "covalant" as "covalent" in two places with "modification".
- Some of the sections in Table 1 were highlighted in yellow. The authors should be more careful to submit a clean version of text for the reviewers and editors.
In view of my comments above, I would recommend a major revision of the manuscript.
Author Response
Comments are covered and highlighted in the manuscript.

Reviewer 3 Report
The revision entitled "Properties and applications of graphene and its derivatives in biosensors for cancer detection: A comprehensive review" submitted by Pourmadadi et al. to biosensors summarizes the relevant advances on graphene materials based biosensors for cancer detection. It is an exciting subject and, therefore, an excellent addition to the literature.
The authors carefully discuss the subject, and I don't have any concerns about its quality and importance. However, a few improvements and clarifications should be made before publication.
For instance, it is possible to find many misspellings and typos throughout the manuscript. Please be careful and correct them. For instance:
Line 40: period should be after the reference. There are others.
Line 61: And should read and
Line 601: miRNA-2 should be miRNA-21
Line 51 and 57: Several double spaces are used throughout the text.
Line 606 superscripts are not in place.
A comparison of the LOD values obtained for more common technologies used (e.g. PCR and ELISA) should be discussed, relating them with the values achieved for the graphene biosensors discussed in this manuscript.
Both tables are numbered table 1. Figure one should have captions with a bigger font. Figure 2 should be replaced with a figure with higher quality.
Also, the conclusion would benefit from a more personalized discussion with future directions on how these biosensors should evolve to become as competitive as standard detection systems already used to detect cancer biomarkers.
Author Response

(The authors gave the same response as above.)

Round 2
Reviewer 1 Report
The authors have done all the changes as per the response to the reviewer.
So the manuscript can be accepted in the present form
Author Response
The last note from the reviewer confirms the manuscript's readiness for publication:
"The authors have done all the changes as per the response to the reviewer.
So the manuscript can be accepted in the present form."
Reviewer 2 Report
The review is comprehensive and interesting. however, there are far too many typos and grammatical errors, thus, an extensive editing of English language and style is absolutely required. I have also listed below some issues that must be addressed in a minor revision:
- On page 9, Figure 5 must be deleted, as this is a very general information that can be written in the text. A dedicated figure for this information is not required.
"Figure 5. chart of modification of graphene by different types of biorecognition elements"
2. On page 8, line 320, the authors wrote:
"Linear (LSV), Differential pulse voltammetry (DPV), Square wave voltammetry (SWV), Stripping voltammetry (SV)..."
these abbreviations must be corrected as:
"Linear sweep voltammetry (LSV), differential pulse voltammetry (DPV), square wave voltammetry (SWV), adsorptive stripping voltammetry (AdSV)..."
3. The citations 177 and 178 were the same. The authors must double-check their references to avoid similar repetitions.
177. Ma, K., et al., Graphene-Oxide Based Fluorescent DNA Aptasensor for Liver Cancer Diagnosis and Therapy. 2021. 1319
178. Ma, K., et al., Graphene Oxide Based Fluorescent DNA Aptasensor for Liver Cancer Diagnosis and Therapy. Advanced Functional 1320 Materials, 2021. 31(36): p. 2102645.
4. Some of the figure caption include the information that the copyright permission was obtained from those references. However, some of the figure captions do not include this information. The authors must confirm that they obtained the appropriate copyright permissions for all the figures.
Author Response
- On page 9, Figure 5 must be deleted, as this is a very general information that can be written in the text. A dedicated figure for this information is not required.
"Figure 5. chart of modification of graphene by different types of biorecognition elements"
- The mentioned figure has been deleted.
2. On page 8, line 320, the authors wrote:
"Linear (LSV), Differential pulse voltammetry (DPV), Square wave voltammetry (SWV), Stripping voltammetry (SV)..."
these abbreviations must be corrected as:
"Linear sweep voltammetry (LSV), differential pulse voltammetry (DPV), square wave voltammetry (SWV), adsorptive stripping voltammetry (AdSV)..."
- Corrected and highlighted in the manuscript.
3. The citations 177 and 178 were the same. The authors must double-check their references to avoid similar repetitions.
177. Ma, K., et al., Graphene-Oxide Based Fluorescent DNA Aptasensor for Liver Cancer Diagnosis and Therapy. 2021. 1319
178. Ma, K., et al., Graphene Oxide Based Fluorescent DNA Aptasensor for Liver Cancer Diagnosis and Therapy. Advanced Functional 1320 Materials, 2021. 31(36): p. 2102645.
- These two references have been merged together.
4. Some of the figure caption include the information that the copyright permission was obtained from those references. However, some of the figure captions do not include this information. The authors must confirm that they obtained the appropriate copyright permissions for all the figures.
- Some figures have been added to the manuscript after the first revision. The permissions will be sent to the journal.